# Evaluation of the 10&10,000 Change Challenge Program

**DOI:** 10.3390/nu17091494

**Published:** 2025-04-29

**Authors:** Megan McClendon Pynckel, Sumathi Venkatesh, Mark D. Faries

**Affiliations:** 1Family & Community Health, Texas A&M AgriLife Extension Service, College Station, TX 77845, USA; mark.faries@ag.tamu.edu; 2Department of Nutrition, Texas A&M AgriLife Extension Service, College Station, TX 77845, USA; sumathi.venkatesh@ag.tamu.edu; 3School of Public Health, Texas A&M Health Science Center, College Station, TX 77845, USA

**Keywords:** lifestyle behaviors, health program, physical activity, diet, nutrition, fruits, vegetables

## Abstract

Background: Chronic disease is the leading cause of death in the United States but can be prevented with lifestyle behaviors, such as physical activity and healthy dietary habits. Experts have developed health programs to promote these behaviors, but they have not led to long-term effects or focused on meeting guidelines and recommendations. Objectives: To determine the effectiveness of a health program in improving participants’ confidence levels, health outcomes, and health behaviors. Methods: Within-subjects analysis was conducted to determine pre-post changes in weight, health, and confidence in achieving step count and fruit and vegetable consumption goals. A between-subjects analysis was conducted to compare program graduates and program dropouts to determine the influence of weight classification, weight, health, and confidence on program completion. Results: After completing the program, program graduates lost an average of six pounds and reported increases in health and confidence in achieving step count and fruit and vegetable goals. Health and confidence levels were significantly lower among program dropouts compared to program graduates. Conclusions: The program had a positive effect on confidence levels and health outcomes among program graduates. Materials to enhance confidence should be considered for distribution during the pre-assessment phase of this program.

## 1. Introduction

Healthy lifestyle behaviors can reduce the risk of leading chronic diseases, including heart disease, hypertension, cancer, and type 2 diabetes [1]. Obesity is a major risk factor for these diet-related chronic diseases [2,3], yet excess weight gain can be prevented or managed by adopting healthy lifestyle behaviors, such as regular physical activity (PA) and increasing daily fruit and vegetable (FV) consumption [4,5,6]. However, less than half (46.9%) of American adults currently meet the recommended aerobic PA guidelines per week (Centers for Disease Control (CDC) [7], and even fewer meet the recommendations for fruit (12.3%) and vegetable (10%) consumption [8].

Recognizing and addressing the barriers and facilitators to achieving adequate FV intake and PA maintenance is essential for preventing chronic disease, excess weight gain, and associated health impacts. Many barriers to PA have been reported, including lack of time, cost, and limited environmental resources [9,10]. Walking can help overcome these barriers. First, walking is a common personal choice and prescription due to its positive perceptions of cost, time, and convenience [11,12]. Additionally, the Physical Activity Guidelines Advisory Committee has stated walking as an intelligible and accessible form of PA for people of different racial and ethnic groups, fitness levels, and ages [13]. Walking that achieves at least 7000–8000 steps per day (and up to 10,000 steps per day) could be a comparable alternative to achieving the recommended 150 min a week of moderate-intensity PA [13,14,15]. Engaging in this level of walking is associated with improvements in blood pressure [16,17], bone health [18], lipid profiles [19,20,21], and body weight [21]. Subsequently, losing 5% or more of body weight is linked to reductions in plasma glucose [22], blood pressure [23], total cholesterol, and cholesterol ratio [24] as well as increases in high-density lipoprotein [25]. This holds true even for children, with improvements seen in triglyceride levels, plasma insulin, and cholesterol ratio [26].

The Dietary Guidelines for Americans 2020–2025 recommend a healthy dietary pattern that prioritizes nutrient-dense food choices, with a particular emphasis on including a variety of FV in fresh, frozen, canned, or dried forms [27]. More specifically, consuming 7 to 7.5 servings of FVs per day is associated with a reduced risk of total cancer, while consuming 10 servings per day is linked to a reduced risk of coronary heart disease, stroke, cardiovascular disease, and all-cause mortality [28]. Yet, despite such recommendations, limited time, affordability concerns, and taste preferences are often cited as barriers to adequate FV intake [29,30].

Various theories have been developed to understand and guide individuals to overcome common barriers to adopting a healthy lifestyle of adequate FV intake and PA levels. Of those, self-regulation is a common concept and includes many behavioral techniques and constructs. Self-regulation is the internalization of self-control and includes cognitive flexibility, a necessary trait for personal growth and development. Self-monitoring, a construct of self-regulation, promotes action beyond mere intention [31] by monitoring behavior and health status, thereby increasing the likelihood of positive health behavior changes. Self-monitoring has improved step count and facilitated weight loss, resulting in sustainable changes in the short and long term as well as improved goal attainment. However, further research is needed as previous programming is sparse with variable outcomes [32].

Of interest here is the goal of early-stage behavior intervention to improve self-efficacy. Self-efficacy is an individual’s belief in his or her ability to complete a task—often reflected as one’s confidence in their ability to exert control over motivation, behavior, and social environment [33,34,35]. Self-efficacy is a dimension of self-regulation, which has a strong association with behavior change among diverse populations [32]. The two primary phases of self-efficacy are motivation and volition. Motivational self-efficacy is essential for task initiation and goal setting, while volitional self-efficacy facilitates the achievement of goals once the behavior has been initiated [36]. Therefore, enhancing self-efficacy could promote health related intentions and health behaviors, especially during the intention formation phase that occurs in the early stages of behavior change [37]. Self-efficacy has been positively associated with dietary and exercise behaviors [38] and is noted as a central mechanism of personal agency that determines self-regulation [39], making it important in both physical activity and nutrition. Self-efficacy has been shown to have a greater impact than one’s perceptions of their food environment and cooking confidence. Studies have documented an increase in self-efficacy for healthy eating in a group of 513 rural women [40] and an improvement in mean FV intake among individuals with low income and educational levels [41]. For physical activity, self-monitoring in the form of barrier identification and action planning has the most significant impact on improving self-efficacy and enhancing physical activity behavior [42].

Despite evidence supporting self-regulatory constructs, there remains a gap within health education and programming to successfully affect early-stage behavior and improve self-efficacy. Therefore, Texas A&M AgriLife Extension Service (AgriLife Extension) [43] developed the 10 & 10,000 Change Challenge program (10&10K), as an individual approach to promote health behavior recommendations and to achieve daily goals of walking 10,000 steps and consuming 10 servings of FVs at least thrice a week in 100 days. Self-regulation tasks, called ‘tip tasks’, were developed to enhance self-monitoring and self-efficacy throughout the program. We chose a 100-day intervention to allow realistic goal setting and to facilitate a gradual increase in step count and servings of FV. This study is an example of early work completed by Marcus and colleagues [44] that show, not only is self-efficacy low in beginning stages of change, but high in later stages.

### The 10 & 10,000 Change Challenge Program

10&10K is a web-based program developed in 2020 by AgriLife Extension. The program was designed to help individuals progress across four stages that incrementally increase daily step count and dietary goals for at least three days a week—Stage 1: 3000 steps and three FVs, Stage 2: 5000 steps and five FVs, Stage 3: 7500 steps and seven FVs, Stage 4: 10,000 and ten FVs. In addition to logging progress during each stage, participants are asked to complete three “tip tasks” that were created and reviewed by content experts. Each tip task involves reviewing a short video on improving self-efficacy by identifying PA and FV barriers and facilitators followed by questions or tasks, that help participants practice the skills necessary to sustain health behavior practices. One tip task, “pay attention”, informs participants that paying attention brings awareness to facilitators and barriers to PA. Prompts for this task included open-ended questions, such as “why did you miss your last workout?” and “why did you make your last workout?” These questions prompt participants to reflect on the factors that led them to either miss or complete their workouts. Tip tasks were created using techniques based on self-determination theory [45] and were designed to promote motivation to reach stage goals through self-monitoring [46]. 

## 2. Methods

### 2.1. Aims

The purpose of this paper is to evaluate the effectiveness of the 10&10K challenge in improving the weight status, general health perception, and self-efficacy of program participants in increasing their FV intake and PA level. The aims were to (1) assess the overall and within-subjects changes in self-efficacy, weight, and general health from baseline to follow-up among participants who completed the program to provide insight into program effectiveness; (2) determine changes in self-efficacy between weight groups identified through calculation of self-reported height and weight to provide insight into the program’s effectiveness for participants within different body mass index (BMI) categories; (3) assess between-subjects differences in self-efficacy, weight, and general health among participants who completed 10&10K and those who did not complete the program. Qualitative analysis of tip task responses is outside the scope of this paper and will be addressed in future writings.

### 2.2. Participants

Participants were predominantly Texas residents who were recruited through web-based advertisement of the program and in-person recruitment conducted by a statewide cohort of Texas A&M AgriLife County Extension Agents who live and work in the communities they serve. Participating adults (≥18 years) self-opted into the program by selecting 10&10K from a list of available programs (howdyhealth.tamu.edu). A total of 1270 participants enrolled into the program, and 213 participants completed both the pre- and post-program assessments (i.e., “program graduates”). A total of 969 participants completed pre-program assessments only; 965 discontinued participation in the program at varying time points over the allotted 100-day program duration (i.e., “program dropouts”) with four removed from analysis due to incomplete or missing demographic and self-efficacy data. Eighty-eight participants were excluded from the analysis due to the absence of any reported information. Descriptive data of self-reported age, sex, and race were collected at the time of enrollment into the program (see Table 1).

### 2.3. Design and Procedures

Pre- and post-measures and evaluation methods were constructed for internal evaluation and program development purposes. Program activities and data collection are ongoing with this dataset being a sample collected from program inception (September 2020) through September 2023. This study was submitted to the Texas A&M University Institutional Review Board who determined that the study was conducted in accordance with the Declaration of Helsinki and did not require informed consent because risk to participants was minimal.

### 2.4. Measures

Pre- and post-program assessments included self-reported body weight (in pounds), self-reported health, and self-efficacy in PA and FV behavior. Self-reported height (in inches) was assessed at pre-program only. Further details are provided below.

#### 2.4.1. Health

Self-reported health included a single item selected from the BRFSS [47], which asked, “would you say that in general your health is?”. Response options were *excellent*, *very good*, *good*, *fair*, *poor*, and *don’t know/not sure* [48]. This item was assessed to measure baseline perceived health for between-subjects to determine influence of health on program completion and to measure within-subjects changes for those who completed the program.

#### 2.4.2. Self-Efficacy

As a construct, self-efficacy is often measured to assess self-confidence or self-belief in the ability to complete a task [49,50]. In the present study, self-efficacy was assessed using a single-item for both PA and FV, respectively, on a five-point scale ranging from *not at all confident* to *very confident*. Specifically, for PA, participants were asked, “how confident are you in your abilities to get at least 7500 steps each day next week?”. For FV, “how confident are you in your abilities to eat at least 5 servings of fruits and vegetables each day next week?”.

### 2.5. Statistical Analysis

To assess the overall and within-subjects changes in self-efficacy, weight, and general health, paired samples t-tests, frequencies, and percentages were examined to determine program effectiveness as measured by changes and statistically significant differences in weight and self-efficacy to meet PA and FV goals. Participants’ BMI was calculated using self-reported height and weight. Participants were categorized as underweight, healthy weight, overweight, and obese based on the BMI categories of the Centers for Disease Control and Prevention [51].

To determine changes in self-efficacy between weight groups, program graduates (n-213) were grouped into overweight (BMI 25.0 to <30.0) and obese (BMI ≥ 30) categories to conduct a 2 × 2 (time by group) repeated-measures ANOVA to assess changes in pre- and post-program weight and self-efficacy between the two weight groups. Only six participants (2.8%) were categorized as normal (BMI 18.5 to <25) or underweight (BMI < 18.5) and were removed from analysis of Aim 2. Alpha-criterion was set at α = 0.05 for all analyses.

For aim 3, of the total participants with baseline data (n = 1182), program dropouts (n = 969), or those who did not complete the program within the 100-day completion goal, were selected as a sub-sample for comparison with program graduates who successfully completed 10&10K (n = 213). T-tests were used to assess differences in self-efficacy, weight, and general health between these two groups.

## 3. Results

### 3.1. Participant Demographics

Participants who completed 10&10K (n = 213) were mostly female (n = 185, 86.9%), ranged in age from 22 to 69 years (M = 50 years), and were obese (BMI, M = 31.9, 53.5%). Most program graduates self-reported their race/ethnicity as non-Hispanic white (n = 150, 70.4%), with Hispanic (n = 33, 15.5%) being the second largest group, and African American (n = 23, 10.8%) being third.

### 3.2. Changes in Self-Efficacy, Weight, and Health Perception of Program Graduates

For program graduates, Table 1 shows changes in self-efficacy, weight, and health perceptions before and after completing the program, and Table 2 shows results from the repeated measures ANOVA. Approximately 23.7% (n = 49) of program graduates lost 5% of their body weight during the program, and 65.2% (n = 135) lost any weight (M = −6.0). These measures were calculated by assessing actual difference per individual and grouping individuals based on percent of body weight loss with 5% used as a cutoff because of known health benefits [22,23,24,25,26]. We did not see a change in general health perception before (Pre Health, M = 3.0) and after (Post Health, M = 3.3) completing the program in our sample. Within-subjects differences were calculated for general health by calculating differences between pre- and post-assessment and then grouping based on change versus no change. Most participants reported their general health as good at pre- (n = 111, 52.1%) and post-assessment (n = 119, 55.9%), with many (n = 133, 62.4%) not reporting a change. Compared to baseline, the proportion of individuals showing an improvement in confidence levels for achieving PA goals (n = 95; 44.8%) and FV goals (n = 111; 52.4%) increased during the follow-up assessment. As with body weight and general health, within-subjects differences were calculated before grouping was completed.

### 3.3. Differences Based on Weight Group and Completion Status

Significant differences were not found between overweight and obese program graduates when analyzing changes in self-efficacy to reach PA and FV goals and were therefore not included in either table. Statistically significant differences were found between those who completed the 10&10K program (n = 213) and those who did not complete the program (n = 969) when analyzing pre-assessment data. Participants who completed the program self-reported higher levels of perceived health status (Pre Health; M = 3.0 vs. M = 2.8), self-efficacy to reach PA goals (Pre-Confidence Feet; M = 3.9 vs. M = 3.5), and self-efficacy to reach FV goals (Pre-Confidence Fork; M = 3.8 vs. M = 3.3) compared to those who did not complete the program. No differences were found between graduates and dropouts for sex, age, or BMI. See Table 1.

## 4. Discussion

The objective of this study was to evaluate the effectiveness of the 10&10,000 Change Challenge program, an educational program aimed to improve the weight status, general health perception, and self-efficacy of program participants by gradually increasing goals for FV intake and PA level and providing tips during each stage. We compared participants’ changes in self-efficacy, weight, and general health perceptions (a) before and after completing the program, (b) between overweight and obese weight categories, and (c) between program graduates and program dropouts.

### 4.1. Program Outcomes and Implications

Program graduates self-reported higher levels of health and self-efficacy for FV and PA goal attainments, and lower weight in pounds when compared to their baseline information. Previous mobile applications targeting behavior change in PA and diet have not shown effectiveness in modifying both behaviors [52] or in achieving positive health outcomes, including weight change [53,54]. Results from other mobile application studies have shown a decrease in daily step count when daily step goals are implemented [55] or a slight increase in step count over an 8-week period [56]. However, these interventions were not successful in modifying dietary and PA habits, even when used along with tips based on habit theory, which is a framework designed to create a habit through the daily repetition of activities to remove the need for conscious effort [57]. This lends support for the current structure of the 10&10K program highlighting reasons for program efficacy. The 10&10K web-based program allows participants to progress at their own pace with the aim to achieve set daily goals on three separate days over a 100-day period. A pilot period of the 10&10K program revealed that participants were most successful at achieving goals through manageable and infrequent bouts that spanned across 100 days, allowing adequate time for behavior change to occur. Time to adopt health behaviors could vary by behavior type, with approximately 91 days to adopt PA behaviors [58]. Use of tips focused on improving self-efficacy was a strength of this application as research has documented increases in self-efficacy to have a modest effect on behavior [37]. Previous applications either did not use self-efficacy or did not measure changes in the construct, despite inclusion of self-monitoring tips and tasks [55,59]. Measurement of self-efficacy in the current study showed an increase over time, validating the effectiveness of the tip tasks used in this program and their role in efficacy of the 10&10K program.

Weight classification did not play a role in self-reported increases in self-efficacy for FV and PA or in self-reported weight loss, lending support for the generalizability of the program. One possible explanation for this observation could be that improvement in self-efficacy has been shown to be associated with weight loss [60], which was seen in both groups. Given that our sample primarily consisted of overweight and obese individuals, this factor may also influence their perception of self-efficacy and their general health at baseline. As rates of obesity and chronic disease continue to rise in this country, it is a growing concern that health program dropout rates remain high among similar populations despite potential program outcomes, including increased fertility [61] and improved glucose, lipids, and blood pressure control [62].

### 4.2. Program Drop Out

Dropout was high among program participants, with self-efficacy being a potential mediating factor. Mean differences for self-efficacy were small between 10&10K program graduates and program dropouts; however, these differences were significant indicating a need to focus on self-efficacy to increase program efficacy. Self-efficacy is a general predictor of health-related behavior (d+ = 0.11), especially among older adults (d+ = 0.53) and those with diagnosed cancer (d+ = 0.33) [37]. Researchers should consider providing education and tip tasks to increase self-efficacy before program commencement. This practice has shown promising outcomes among female high school students for career decision and STEM participation [63], improving PA among patients with musculoskeletal disorders [64], and among preservice teachers [65]. Each group increased self-efficacy through mastery experience, which is posited to drive efficacy and performance outcomes [66]. Therefore, this approach should be considered to increase self-efficacy before participants enter the 10&10K program.

### 4.3. Strengths and Limitations

The 10&10K program was developed through use of theoretical constructs that have proven effective in previous lifestyle change interventions. Participants completed the program online and had access to county- and state-level assistance for programmatic and technical concerns and questions. Evaluation of this program serves as a strength and serves as a part of the iterative process to increase participant outcomes and program efficacy. Limitations include inability to record influences external to the program and how they may have affected program outcomes, including dietary intake beyond fruit and vegetable consumption. Evaluation measures purposefully excluded such assessment items, as they are outside the scope of organizational measurement capabilities; however, organizations with greater capacity should consider conducting experimental research using food diaries or meal prescription plans. We used self-reported height and weight to calculate participants’ BMI, which could have led to a potential reporting bias and inaccuracies, leading to over or underreporting of weight status. Also, we did not have a sizeable number of participants within the healthy weight range and were unable to make a comparison between overweight and obese groups with those with a BMI of under 25. Since the 10&10K is a web-based program, our sample was restricted to participants with computer access and who were able to effectively use web-based technology. However, to address these potential digital literacy barriers, we included clear instructions with user-friendly navigation and had extension educators who provided technological support.

## 5. Conclusions

This study was completed for internal evaluation and program development purposes. Findings will be used to strengthen programmatic components to increase program efficacy. Self-reported weight, health, and self-efficacy to complete FV and PA goals improved for participants who completed the 10&10K program, revealing that self-efficacy can be affected in 100 days and impact achievement of FV and PA goal achievement. Future research and programming should include preliminary education to improve self-efficacy for FV and PA for participants who self-report low self-efficacy during pre-assessment measures.

## Figures and Tables

**Table 1 nutrients-17-01494-t001:** Descriptive data of programmatic participants.

Measured Parameters	Graduates (N = 213)	Dropouts (N = 965)	t (*p*-Value)
**Sex** (%) Female Male	86.9 13.1	86.7 13.3	
**Age** (M, SD; years)	50.0 (9.8)	48.0 (12.8)	
**Pre-BMI** (M, SD)	31.9 (7.3)	32.8 (11.6)	
**Post-BMI** (M, SD)	30.8 (7.0)		
**Pre-Weight** (M, SD; lbs)	196.6 (45.9)	198.5 (88.5)	−0.3 (0.76)
**Post-Weight** (M, SD; lbs)	189.8 (44.3)		
**Pre-Health** (M, SD)	3.0 (0.9)	2.8 (0.9)	2.1 (0.04) *
**Post-Health** (M, SD)	3.3 (0.8)		
**Pre-Confidence** (M, SD) Feet Fork	3.9 (1.1) 3.8 (1.0)	3.5 (1.3) 3.3 (1.2)	4.6 (0.00) * 6.3 (0.00) *
**Post-Confidence** (M, SD) Feet Fork	4.4 (0.9) 4.5 (0.7)		

Note: M = mean; SD = standard deviation; lbs = pounds; Pre = assessment completed at the beginning of the program; Post = assessment completed at the end of the program; Feet = confidence in ability to achieve step goal; Fork = confidence in ability to achieve FV goal; Post data not applicable for program dropouts; t = differences between Graduates and Dropouts; * = *p* < 0.05 between graduates and dropouts.

**Table 2 nutrients-17-01494-t002:** Paired-samples test among program graduates.

Measured Parameters	Mean Difference (SD)	t	*p*-Value
Weight (lbs; M, SD)	−6.0 (14.2)	−6.0	<0.001
Health (M, SD)	0.3 (0.7)	5.9	<0.001
Confidence (M, SD) Feet Fork	0.5 (1.1) 0.7 (1.1)	6.3 9.4	<0.001 <0.001

Note: SD = standard deviation; t = outcome from repeated measures ANOVA conducted on program graduates; *p*-Value = significance from repeated measures ANOVA test.

## Data Availability

The datasets used and/or analyzed during the current study are available from the corresponding author on reasonable request.

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
