# Peer review of "Evaluation of the 10&10,000 Change Challenge Program"

_nutrients, 2025, doi:10.3390/nu17091494_

Round 1

Reviewer 1 Report

Comments and Suggestions for Authors

The authors conducted a within-subjects analysis to examine pre–post changes in weight, perceived health, and confidence in achieving step count and fruit and vegetable consumption goals. A between-subjects analysis was also performed to compare program graduates (N = 213) and dropouts (N = 965) in order to assess the influence of weight classification, weight, perceived health, and confidence on program completion.

Here are some of my suggestions:

1.

In Section 2.2 Participants, it is stated that a total of 969 participants completed pre-program assessments only. However, Table 1 indicates that the number of dropouts is N = 965. It is unclear whether this discrepancy is due to a typographical error or another issue. This should be addressed and clarified to ensure consistency and transparency in the reporting of participant numbers.

2.

The Results section may be difficult for readers to follow, as key findings are not clearly presented or appropriately referenced. For instance, the statement “Participants who completed the program self-reported higher levels of health (M = 3.0 vs. M = 2.8), self-efficacy to reach PA goals (M = 3.9 vs. M = 3.5), and self-efficacy to reach FV goals (M = 3.8 vs. M = 3.3) compared to those who did not complete the program” should be supported by the corresponding analysis results presented in Table 1 to improve clarity and ensure consistency between the text and tables.

3.

The “Table 2. Paired Samples Test Among Program Graduates” is unclear regarding the specific statistical test used and how the results should be interpreted. Additionally, the statistical method employed should be clearly described, either in the table footnote or in the Methods section, to enhance transparency and reproducibility.

4.

In Section 2.5 Statistical Analysis, it is stated that a 2×2 (time by group) repeated-measures ANOVA was conducted to assess changes in pre- and post-program weight and self-efficacy between overweight (BMI 25.0–<30.0) and obese (BMI >30) program graduates (n = 213). However, it is unclear whether the results of this repeated-measures ANOVA are actually presented in any of the tables or described in the Results section. This lack of clarity may confuse readers, and the description of the statistical analysis should be aligned with the reported results to ensure transparency and interpretability.

5.

It is recommended that the authors provide a more comprehensive description of the descriptive statistics, including a detailed comparison of program graduates (N = 213) and dropouts (N = 965). Additionally, a test for differences between these groups should be included. For the within-subjects analysis (presumably for graduates, N = 213), it would be helpful to present the results of the pre–post changes in the tables. Furthermore, the results of the repeated-measures ANOVA should also be clearly presented in the tables, including the corresponding p-values, to ensure transparency and ease of interpretation.

Author Response

Thank you for your feedback. Please see the attachment with my responses.

Reviewer 2 Report

Comments and Suggestions for Authors

Dear Authors,

Thank you for the opportunity to review your manuscript entitled “Evaluation of the 10&10,000 Change Challenge Program”. It is a well-written and insightful paper. The English is clear and the content is highly relevant. I appreciated the practical approach and the structured analysis. Below are my comments and suggestions for minor revisions:

Introduction

Consider adding a table to summarize the four stages of the 10&10K program. This could improve readability and immediate understanding of the progression logic.

Materials and Methods

A table listing the exact items/questions used to assess self-efficacy, health perception, and dietary behavior could help clarify the methodology.

Results

In Table 1, the line "SD= standard deviation" is missing a space after "SD".

Consider labeling the first column of Table 1, for example as “Measured Parameters” or “Measured Characteristics”.

Line 233: It may be helpful to rephrase “higher levels of health” as “higher levels of perceived health status” or “parameter health” to avoid ambiguity with general health concepts.

In Table 2, consider adding headers to the columns to improve clarity.

Discussion

In the limitations section, consider mentioning that the study did not monitor participants' full dietary intake beyond fruit and vegetable servings. It may be useful to suggest including food diaries or standardized dietary prescriptions (e.g., a Mediterranean diet plan personalized by a nutrition professional) in future iterations of the program.

Line 117: Consider spelling out the number “3” as “three” for consistency with academic style guidelines.

References and Formatting

Please check reference formatting in the text: for example, in line 33, [4, 5, 6] should be [4–6]; also review line 48 and 80.

The template text before Reference 1 should be removed. The list should begin directly with Reference 1.

Formatting/Clean-up

Please remove extra blank lines at: lines 37, 55, 64, 97, 216, 238, and 239.

These are all minor suggestions to improve clarity and consistency. I believe your work provides a valuable contribution to community-based health promotion.

Best regards

Author Response

(The authors gave the same response as above.)
